# Compound Fault Diagnosis of Wind Turbine Gearbox via Modified Signal Quality Coefficient and Versatile Residual Shrinkage Network

**DOI:** 10.3390/s25030913

**Published:** 2025-02-03

**Authors:** Weixiong Jiang, Guanhui Zhao, Zhan Gao, Yuanhang Wang, Jun Wu

**Affiliations:** 1School of Naval Architecture and Ocean Engineering, Huazhong University of Science and Technology, Wuhan 430074, China; jiangweixiong@hust.edu.cn (W.J.); z_gao@hust.edu.cn (Z.G.); 2College of Computer Science and Technology, Zhejiang University, Hangzhou 310058, China; 12321305@zju.edu.cn; 3Sino-German College of Intelligent Manufacturing, Shenzhen Technology University, Shenzhen 518118, China

**Keywords:** compound fault diagnosis, versatile residual shrinkage network, modified signal quality coefficient, wind turbine gearbox

## Abstract

Wind turbine gearbox fault diagnosis is critical to guarantee working efficiency and operational safety. However, the current diagnostic methods face enormous restrictions in handling nonlinear noise signals and intricate compound fault patterns. Herein, a compound fault diagnosis method based on modified signal quality coefficient (MSQC) and versatile residual shrinkage network (VRSN) is proposed to resolve these issues. In detail, the MSQC is designed to remove the noise components irrelevant to wind turbine operation status, and it has the ability to balance the denoised effect and signal fidelity. The VRSN is constructed for compound fault diagnosis, and it consists of two heterogeneous residual shrinkage networks. The former is designed to count the number of faults, and the latter is adopted to identify the single or compound fault pattern. Finally, a self-built wind turbine gearbox compound fault test rig is adopted to verify the proposed method’s effectiveness. The results demonstrate that the proposed method is competitive in terms of compound fault diagnosis accuracy.

## 1. Introduction

Due to high energy transmission efficiency and strong power output, wind turbines are of great importance when used in distributed generator infrastructures [1,2,3]. However, due to the intricate internal structure and harsh working environment, the malfunctions are prone to occur in the key component of a wind turbine. When 6312 wind turbines are surveyed, the distributed reliability working group of the institute of electrical and electronics engineering (DRWG-IEEE) reports that half of all faults derive from the wind turbine gearbox, such as lacking teeth, broken teeth, and bearing inner ring wear. Different faults may even happen at once [4,5,6]. Thus, it is an urge task to develop a fault diagnosis method for wind turbine gearboxes, especially for compound faults.

Many academics have made great efforts to develop a binary combination strategy and a probabilistic-based network for compound fault diagnosis for decades. In the binary combination strategy, multiple binary classifiers are integrated under the 1-versus-1 or 1-versus-all tactics. *m*(*m* − 1)/2 groups of binary classifiers produced by the 1-versus-1 tactic or *m* groups of binary classifiers produced by the 1-versus-all tactic are adopted for the *m*-label classification problem [7]. However, there are several glaring limitations preventing the implementation of the combination strategy, such as the huge computation cost and complex fine-tuning process. In a probabilistic-based network, the Bayesian network is used to compute the fault probability distribution for the observed machine. However, the network decision thresholds and standard training data must be prepared with human intervention in advance, thus they are impractical in real-world applications [8].

Recently, algorithm adaptation methods have been explored to address the above issues. Clare et al. proposed the multilabel decision tree method based on a multilabel entropy and decision tree to realize multilabel compound fault diagnosis [9]. Zhang et al. constructed the multiclass *k* nearest neighbor model based on the *k* nearest algorithm and maximum posterior theory [10]. Tahir et al. established the rank support vector machine based on the maximum margin theory to update a set of linear classifiers, which can handle the multiclass nonlinear problem when the empirical rank error is at a minimum [11]. Liu et al. constructed a classifier chain to explore the label’s interior relationship, but this fails to implement multithread operation because of its chain structure [12]. Wang et al. refined the label power-set by use of random *k* labelsets to propose the random *k* labelsets (RAKEL) for classification efficiency improvement. However, the coupling of a homogeneous-component multilabel classifier in RAKEL may impact the classification performance [13]. Thus, the unreasonable model structure and computational resource configuration will lead to inferior diagnosistic accuracy and efficiency.

On the other hand, the original collected signals are usually nonstationary, nonlinear, and prone to be disturbed by the environment background noises. It is crucial for the preprocessing technique to remove noise components to reveal fault-related features. The current signal preprocessing methods mainly focus on multimodal signal fusion [14,15,16], high-resolution signal decomposition [17,18,19], and end-to-end feature extraction techniques [20,21,22]. However, it is still intractable to process the nonlinear noisy data by use of the traditional denoising technique. In addition, diverse fault types and compound fault patterns further intensify the challenges of signal preprocessing.

To resolve the limitations of the above methods, the compound fault diagnosis method is proposed based on the modified signal quality coefficient (MSQC) and versatile residual shrinkage network (VRSN) for a wind turbine gearbox. The MSQC is designed to detect and remove the noise components irrelevant to the wind turbine’s operation status. Then, the VRSN is established for compound fault diagnosis, and it consists of two heterogeneous residual shrinkage networks used for the fault count and fault probability distribution calculation. The main contributions of the paper are as follows:(1)The VRSN is proposed to diagnose compound faults in a wind turbine gearbox. Different from the probabilistic-based method, the proposed network is self-adaptive, and can identify single or simultaneous faults without manual intervention for empirical threshold setting;(2)The multithread network structure is constructed to optimize the computation resources’ configuration. Two parallel residual shrinkage networks can be implemented simultaneously to count the fault numbers and determine the fault probability distribution in responding to the real-time fault diagnosis task;(3)The denoised algorithm is designed to remove the noise components irrelevant to wind turbine operation status. The modified signal quality coefficient has the ability to balance the denoised effect and signal fidelity, and fault-sensitive features hidden in the originally collected signals can be captured precisely.

The remaining of this paper is organized as follows. The basic principle is introduced in Section 2. The proposed method is elaborated in Section 3. The self-built wind turbine gearbox compound fault simulation test rig is described in Section 4. The denoised and diagnostic results are discussed in Section 5. Eventually, conclusions are drawn in Section 6.

## 2. Theoretical Background

### 2.1. Deep Residual Shrinkage Network

The residual network (ResNet) has an excellent classification ability because it can avoid the vanishing gradient and overfitting phenomena produced by model error backpropagation in the identical path. Based on this, the deep residual shrinkage network (DRSN) was proposed, introducing the soft threshold function and attention mechanism into the ResNet. It can adaptively eliminate the noise-related features to improve the model classification’s performance. The soft threshold function is added to eliminate the noise-related data further, as follows:(1)$$f(x_{de})=\left\{\begin{array}{cc} x_{de}-\tau , & x_{de}>\tau \\ 0, & -\tau \leq x_{de}\leq \tau \\ x_{de}+\tau , & x_{de}<\tau \end{array}\right.$$
where *x_de_* is the denoised data, $f(x_{de})$ is the processed feature, and τ is the soft threshold, which can keep the prominent data stable and transform noise-related data to zero. The soft threshold is adaptively calculated by the residual shrinkage building unit (RSBU) [23].

The Sigmoid function is embedded at the end of RSUB as the output layer of DRSN, as(2)$$\alpha_{C}=\frac{1}{1+e^{-z_{C}}}$$
where $Z_{c}$ is the output of the fully connected layer, and $\alpha_{C}$ is the channel scaling value. The channel soft threshold is determined as(3)$$\tau_{C}=\alpha_{C}\cdot \underset{w,h}{\mathrm{average}}\left|x_{W,H,C}\right|$$
where *W*, *H*, and *C* denote the width, height, and channel indexes of the feature map *x*.

### 2.2. ICEEMDAN

The improved complete ensemble empirical mode decomposition with adaptive noise (ICEEMDAN), which is the extended version, can address the residual noise and spurious mode problems that arise with the CEEMDAN algorithm [24]. The principle of the ICEEMDAN algorithm is elaborated in the following.

The Gaussian white noise $w^{\left(i\right)}$ is decomposed by the empirical mode decomposition (EMD), and it is added into the raw signal $x^{\mathrm{raw}}$ to construct the series.(4)$$x^{\left(i\right)}=x+\beta_{\left(i-1\right)}E\left(w^{\left(i\right)}\right),\;i=1,2,\ldots ,I$$
where *x* is the raw signal, $x^{\left(i\right)}$ is the constructed series, $E_{k}\left(\cdot \right)$ is the *k*th order intrinsic mode function (IMF) decomposed by the EMD algorithm, and $\beta_{\left(i-1\right)}>0$.

The first residual component is calculated, and the first mode is given as(5)$$d_{1}=x-R_{1}=x-\left(N\left(x^{\left(i\right)}\right)\right)$$
where $d_{1}$ is the first mode, $R_{1}$ is the first residual component, and $N\left(\cdot \right)$ is the local mean.

The Gaussian white noise is added again. The second residual component is calculated, and the second mode is determined as(6)$$d_{2}=R_{1}-R_{2}=R_{1}-\left(N\left(R_{1}+\beta_{1}E\left(w^{\left(i\right)}\right)\right)\right)$$
where $d_{2}$ is the second mode and $R_{2}$ is the second residual component.

The *k*th residual component is calculated, and the *k*th mode is expressed as(7)$$d_{k}=R_{k-1}-R_{k}=R_{k-1}-\left(N\left(R_{k-1}+\beta_{k-1}E\left(w^{\left(i\right)}\right)\right)\right)$$

Finally, all modes and residual components are aggregated into the reconstructed signal as(8)$$x_{a}\left(t\right)=\sum_{i=1}^{k}d_{i}+R_{k},i=1,2,\ldots ,k$$

## 3. Methodology

### 3.1. Modified Signal Quality Coefficient

The modified signal quality coefficient (MSQC) is designed to balance the signal fidelity and noise reduction, as exhibited in Figure 1. The specific processes are elaborated as follows.

Step 1: The ICEEMDAN is adopted to calculate the IMF and residual components of the raw signal.

Step 2: The effective IMF component number *NE* is determined from 1 to *k*/2, and it depends on the raw signal complexity [25].

Step 3: The Pearson correlation coefficient $R_{i}$ and kurtosis index $K_{i}$ between the effective IMF components and raw signal are calculated. The impulse signal contained in heavy background noise can be detected by the kurtosis index, and the relevance between the raw signal and the effective IMF components can be reflected by the Pearson correlation coefficient. The above indicators are adopted to remove the irrelevant components.
(9)$$R_{i}=\frac{\sum_{t=1}^{T}\left(d_{k}(t)-\bar{d_{k}(t)}\right)\left(x^{\left(i\right)}(t)-\bar{x^{\left(i\right)}(t)}\right)}{\sqrt{\sum_{t=1}^{T}{\left(d_{k}(t)-\bar{d_{k}(t)}\right)}^{2}{\left(x^{\left(i\right)}(t)-\bar{x^{\left(i\right)}(t)}\right)}^{2}}}$$
(10)$$K_{i}=\frac{T\sum_{t=1}^{T}{\left({\mathrm{IMF}}_{i}(t)-\bar{{\mathrm{IMF}}_{i}(t)}\right)}^{4}}{{\left(\sum_{t=1}^{T}{\left({\mathrm{IMF}}_{i}(t)-\bar{{\mathrm{IMF}}_{i}(t)}\right)}^{2}\right)}^{2}}$$where $\bar{d_{k}(t)}$ and $\bar{x^{\left(i\right)}(t)}$ are the averages of $d_{k}(t)$ and $x^{\left(i\right)}(t)$.

Step 4: After selecting the effective IMF components, the reconstructed signal $x_{bg}^{lz,(i)}(t)$ and the IMF number record matrix $xm_{de}^{lz,(i)}(t)$ are determined, where *l* is the IMF number in $x_{de}^{lz,(i)}(t)$, *z* is the reconstructed signal number, *b* is the first *b* IMFs selected by the amplitude of $R_{i}$, *g* is the first *g* IMFs selected by the amplitude of $K_{i}$, the IMF represented by *b* cannot overlap with the IMF represented by *g*, and *l* = *b* + *g*. The Pearson correlation coefficients represent a prioritized index. For example, when *M* = 1, 2, and 3, the IMF selection rule is as shown in Figure 2.

When *M* = 1, the reconstructed signal $x_{10}^{11}$ and the record matrix of IMF order $xm_{10}^{11}$ are acquired. When *M* = 2, the reconstructed signal $x_{20}^{21}$, $x_{11}^{22}$ and the record matrix of IMF order $xm_{20}^{21}$, $xm_{11}^{22}$ are acquired. When *M* = 3, the reconstructed signals $x_{30}^{31}$, $x_{21}^{32}$, $x_{21}^{33}$ and $x_{21}^{34}$ and the record matrices of IMF orders $xm_{30}^{31}$, $xm_{21}^{32}$, $xm_{21}^{33}$ and $xm_{12}^{34}$ are acquired.

Step 5: The Pearson correlation coefficient $R_{x_{de}^{lz,\left(i\right)}}$, mean square error $MSE_{x_{bg}^{lz,(i)}}$, and redefined signal-to-noise ratio $MSNR_{x_{bg}^{lz,(i)}}$ between the reconstructed signal and the raw signal are calculated. (11)$$R_{x_{bg}^{lz,(i)}}=\frac{\sum_{t=1}^{T}\left(x_{bg}^{lz,(i)}(t)-\bar{x_{bg}^{lz,(i)}(t)}\right)\left(x_{a}(t)-\bar{x_{a}(t)}\right)}{\sqrt{\sum_{t=1}^{T}{\left(x_{bg}^{lz,(i)}(t)-\bar{x_{bg}^{lz,(i)}(t)}\right)}^{2}{\left(x_{a}(t)-\bar{x_{a}(t)}\right)}^{2}}}$$
(12)$$MSE_{x_{bg}^{lz,(i)}}=\frac{1}{T}\sum_{t=1}^{T}{\left(x_{a}(t)-x_{bg}^{lz,(i)}(t)\right)}^{2}$$
(13)$$MSNR_{x_{bg}^{lz,(i)}}=10\times \log_{10}\left(\frac{\sum_{t=1}^{T}{\left(x_{a}(t)\right)}^{2}}{\sum_{t=1}^{T}{\left(x_{a}(t)-x_{bg}^{lz,(i)}(t)\right)}^{2}}\right)$$

The $R_{x_{bg}^{lz,(i)}}$ can guarantee the signal fidelity, and the larger value indicates that more fault information is contained in the reconstructed signal. The $MSE_{x_{bg}^{lz,(i)}}$ can represent the approximation of the reconstructed signal to the raw signal. The lower its value, the better the denoised performance. The $MSNR_{x_{bg}^{lz,(i)}}$ can reflect the energy proportion of the used signal to the noise signal. The larger its value, the lower the noise proportion in raw signals.

Step 6: The modified signal quality coefficient is defined as follows.(14)$$MSQC_{x_{bg}^{lz,(i)}}=NR_{x_{bg}^{lz,(i)}}+NMSNR_{x_{bg}^{lz,(i)}}+1/NMSE_{x_{bg}^{lz,(i)}}$$
where $NR_{x_{bg}^{lz,(i)}}$, $NMSE_{x_{bg}^{lz,(i)}}$, and $NMSNR_{x_{bg}^{lz,(i)}}$ are the normalization values of the Pearson correlation coefficient, mean square error and modified signal-to-noise ratio. The optimal denoised signal $x_{de}$ can be determined by reconstructing *NE* effective IMF components with the maximal $MSQC_{x_{bg}^{lz,(i)}}$. The flowchart of overall signal denoising processes based on MSQC can be seen in Figure 3.

### 3.2. Versatile Residual Shrinkage Network

The versatile residual shrinkage network (VRSN) consists of Counter-DRSN and Locator-DRSN, and the architecture of VRSN is exhibited in Figure 4.

#### 3.2.1. Counter-DRSN

This network aims to count the fault number. Firstly, the original signals *x_s_* are processed by the MSQC, and the denoised signals *x_de_* with corresponding fault number labels $M_{s}\in \left(1,\ldots ,W\right)$ are fed into Counter-DRSN. The predicted fault number is output as follows.
(15)$$\hat{M}=softmax\left(\sum_{p,q=1}^{P,Q}W_{pq}^{l,C}f\left(x_{de}\right)+b_{q}^{l,C}\right)$$where $\hat{M}$ is the predicted fault number, *s* is the sample number, $W_{pq}^{l,C}$ is the weight vector of Counter-DRSN between the *p*th neural at the *l*th layer and the *q*th neural at the (*l* + 1)th layer, $x_{de}$ is the denoised signal, and $b_{q}^{l,C}$ is the Counter-DRSN biases of all the *l*th layer’s neurons for the (*l* + 1)th neural.

The objective function of Counter-DRSN is given by(16)$$Obj=\frac{1}{s}{\sum_{i=1}^{s}\left(M_{i}-{\hat{M}}_{i}\right)}^{2}$$
where Obj is the objective function.

#### 3.2.2. Locator-DRSN

The purpose of this network is to calculate the fault probability distribution and determine the fault pattern. Similarly, the original signals are denoised by the MSQC. The denoised signals with corresponding fault category labels $U_{s}\in \left(1,\ldots ,C\right)$ are input into Locator-DRSN, and the soft threshold function is added to eliminate the noise-relative components. The fault probability distribution is output from the fully connected layer of the Locator-DRSN as (17)$$\hat{U}=SoftProb\left(\sum_{p,q=1}^{P,Q}W_{pq}^{l,L}f\left(x_{de}\right)+b_{q}^{l,L}\right)$$where $\hat{U}$ is the predicted fault probability distribution, $W_{pq}^{l,L}$ is the weight vector of Locator-DRSN, and $b_{q}^{l,L}$ is the network bias.

Then, the Locator-DRSN is updated by the objective function as follows (18)$$l_{L}=-\frac{1}{S}\sum_{i=1}^{S}\sum_{j=1}^{C}E_{ij}\log\left({\hat{U}}_{ij}\right)$$where $E_{ij}$ is one if the sample *i* belongs to fault category *j*, and zero otherwise. ${\hat{U}}_{ij}$ is the occurrence probability of the *j*th fault category.

Finally, the predicted fault number and probability distribution are aggregated to determine the fault pattern as(19)$$\underset{\left\{i=1,\ldots ,\hat{M}\right\}}{LABEL_{i}}\left(x_{de}\right)=\left\{c_{i}|\arg\max\underset{\left\{c=1,\ldots ,C\right\}}{\left\{\hat{M},\ldots ,0\right\}}\hat{U}(x_{de})\right\}$$
where $LABEL_{i}\left(x_{de}\right)\in \left(c_{1},\ldots ,c_{\hat{M}}\right)$ is the predicted label of the VRSN.

## 4. Experimental Study

To verify the proposed diagnosis method, the self-built wind turbine gearbox compound fault test platform was adopted to place the sample under single and compound fault patterns [26]. This is composed of a multistage gearbox, a generator motor, a prime mover, an electric load simulator, a data collection device, and a laptop, as shown in Figure 5. The rotational speed of a prime mover is 1400 RPM, and two meshing gearsets rotate at 1184 and 840 RPM. One health condition (H1), five single fault patterns (SFP1-SFP5) and six compound fault patterns (CFP1-CFP6) are simulated as exhibited in Figure 5 and Table 1. The fault units are processed artificially. For instance, the broken tooth is manufactured by the laser cutting method to cut a part of a gear tooth.

The raw vibration signal was acquired by use of the triaxial accelerometer (Type, NI-cDaq-9174; sensitivity, 100 mV/g). Each sample consisted of 2048 sampling points (2 s × 1024 Hz). There were 1500 samples under each fault pattern, and they were divided into 70%, 10% and 20% for the model training, validation and testing, respectively. The proposed algorithm was executed with MATLAB R2022b and Python v3.8, and conducted on a personal laptop with Intel Core i7 Processor 14900HK CPU, 32 GB memory, and a Microsoft Windows 11 enterprise operation system. The overall schematic of the experimental study design can be seen in Figure 6.

## 5. Results and Discussion

### 5.1. Signal Denoised Performance

The denoised results of raw vibration signals under H1, SFP1, SFP2, and CFP1 are shown in Figure 7. In Figure 7a,b, the amplitudes of vibration signals under SFP1, SFP2, and CFP1 are obviously larger than that under H1. This phenomenon helps us to identify different fault patterns from aspects of time-domain discrepancy. In Figure 7c, the MSQCs of each IMF component under four fault patterns are exhibited. When the MSQC threshold is 0.25, the denoised signals are obtained by reconstructing the effective IMF components exceeding the MSQC threshold. This indicates that the no. 3, no. 5, and no. 7–9 IMFs are reserved under SFP1, SFP2, and CFP1, but the no. 3 IMF is eliminated, and no. 4 IMF is reserved under H1.

To achieve the optimal denoised effect, the denoised results of CFP1 with MSQC threshold = 0.25, 0.50, and 0.75 are exhibited in Figure 8. The waveforms of raw vibration and denoised signals show no obvious differences, and the frequency components are also similar. The influence of irrelevant noise components cannot be eliminated with the lower MSQC threshold. On the contrary, only 80–200 Hz frequency components are left, and the amplitude of denoised signals decreases significantly. The higher MSQC threshold eliminates the plentiful frequency components relevant to the wind turbine operation condition, and it will bring about a negative impact on the fault identification. Therefore, an appropriate threshold is critical for the denoised performance, and it is set to 0.5 through the repeated experimentation and signal analyses, as shown in Figure 8b.

The denoised algorithm is used to process the raw vibration signals under CFP1, as shown in Figure 9. The MSQCs of no. 3, no. 5, and no. 7–9 IMFs exceed the threshold due to the lower mean square error, modified signal-to-noise, and larger correlation coefficient, as exhibited in Figure 9a. The reserved IMFs have plentiful wind turbine operation status information, and they are integrated to obtain the denoised signals. In Figure 9b, the waveforms of denoised signals present obvious periodicity, and their amplitude fluctuation is more subtle than that of the raw vibration signal.

The redundant IMF components and residuals are removed, and the effective IMFs are left to reconstruct the denoised signals. The original and denoised signals under H1, IFP1, IFP2, and CFP1 are shown in Figure 10. The influence of fault occurrence is more prominent on the waveform and periodicity of denoised signals, and the average of denoised signals decreases obviously. In addition, the irrelevant frequency components are eliminated according to the wind turbine operation status and fault characteristic, and the distribution of denoised signal energy is more centralized. The raw vibration signals under twelve fault patterns are processed as mentioned above, and this will help to enhance the performance of the proposed fault diagnosis method.

### 5.2. Diagnosis Result and Discussion

The repetition experiments were conducted to evaluate the performance stability of the compound fault diagnosis method. In Figure 11, the diagnosis accuracies of 15 repetition experiments are recorded. The overall fault diagnosis accuracies for the test sets are steady, and the mean value is up to 96.16%. Compared with the model training approach using raw vibration signals, the mean value of diagnosis accuracy increases by 4.71% (from 91.45% to 96.16%). This indicates that the denoised algorithm can eliminate irrelevant noise interference and improve diagnostic performance effectively.

The diagnosis results after 15 repetition experiments are shown in Figure 12. In Figure 12a, the average fault probability distribution and fault number are recorded. The predicted fault numbers show larger fluctuation under IFP3 and CFP5. Their test accuracies are relatively lower. However, the propose compound diagnosis method can still identify fault patterns precisely, as exhibited in Figure 12b, and only some of the test samples are misjudged, which were identified as BT/CT under IFP3 and CL/CT under CFP5.

The determination of the deep neural network parameter is still a great problem. There are no mature theories to guide the process of parameter setup [27]. Herin, the hyper parameters of the proposed network such as batch size, learning rate, hidden unit, and the dropout rate are determined by means of the cross-validation and repeated experience [28].

In Figure 13, the relation between the parameter group and the testing loss is exhibited for the last trial. In detail, the candidates of batch size, learning rate, dropout rate, and hidden unit are [2, 4, 8, 16, 32, 64], [1 × 10^−6^, 1 × 10^−5^, 1 × 10^−4^, 1 × 10^−3^, 1 × 10^−2^, 1 × 10^−1^], [0.1, 0.2, 0.3, 0.4, 0.5, 0.6, 0.7], and [100, 200, 300, 400, 500, 600, 700, 800, 900]. The optimal batch size, learning rate, dropout rate, and hidden unit of Counter-DRSN are 64, 1 × 10^−4^, 0.1, and 900. Those of the Locator-DRSN are 64, 1 × 10^−4^, 0.4, and 500.

### 5.3. Comparison Analysis

To verify the effectiveness of the proposed method, there are five compound fault diagnosis algorithms used for the comparison analysis, including pairwise probabilistic multilabel classification (PPMLC) [8], random *k* labelsets (RAKEL) [13], dual-extreme learning machine (Dual-ELM) [6], and wavelet transform multi-label convolutional neural network (WT-MLCNN) [27]. The denoised algorithm was here applied in the comparative algorithms for further denoised effect analysis. To eliminate the contingency in the results, twenty repeated experiments were conducted, and the same samples were used for the model training, validation, and testing. The comparison analysis results are recorded in Table 2.

The average test accuracy of the proposed compound method reached up to 96.26%, followed by method 8 (95.83%), method 7 (92.72%), method 2 (88.32%), method 1 (86.03%), and so on. It is obvious that after applying the denoised algorithm, the average accuracies increased by about 4% for four comparative algorithms. However, the standard deviations of methods 3, 4, and 5 were still larger, and thus showed inferior performance stability. To improve the efficiency of diagnosis, method 5 only required 21.75 s to execute the diagnostic tasks, followed by method 6 (39.57 s), method 9 (52.34 s), method 1 (57.95 s), and so on. The execution time for the proposed method is close to those of the first two machine learning-based methods. The comparison analysis results demonstrate that the proposed method is superior in terms of diagnosis accuracy and efficiency.

## 6. Conclusions

The compound fault diagnosis method is proposed based on the modified signal quality coefficient and versatile residual shrinkage network. The main conclusions are summarized as follows:1.The signal denoised algorithm is designed to remove the noise components irrelevant to wind turbine operation status. In this paper, the modified signal quality coefficient can balance the denoised effect and signal fidelity, and fault-sensitive features hidden in the original collected signals can be excavated precisely;2.A versatile residual shrinkage network is constructed for the compound fault diagnosis. Unlike the probabilistic-based method, the proposed network is self-adaptive, and is used to identify single- or simultaneous-fault scenarios without the manual intervention required for setting the empirical threshold;3.An effective multithread network structure is constructed to optimize the computation resource configuration. Two parallel residual shrinkage networks can be implemented simultaneously to count the number of faults and determine the fault probability distribution used for responding to a real-time fault diagnosis task.

In future research, physical interpretability methods such as the physics-knowledge-guided method can be employed for the construction of neural network because they show good performance in improving the stability of neural networks and enhancing the interpretability of the diagnostic processes [29]. In addition, different fault categories entail different degrees of risk regarding the health of a wind turbine, and the fault risk weight can be evaluated by use of the multi-criteria decision-making method, as an extension of the current research topic.

## Figures and Tables

**Figure 1 sensors-25-00913-f001:**
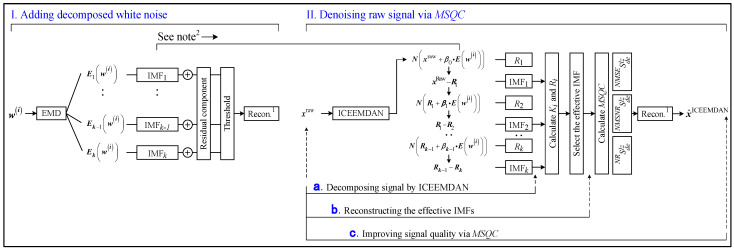
Methodology of modified signal quality coefficient. (Notes: ^1^ reconstruction; ^2^ decomposing Gaussian white noise by EMD, and adding it into the raw signal).

**Figure 2 sensors-25-00913-f002:**
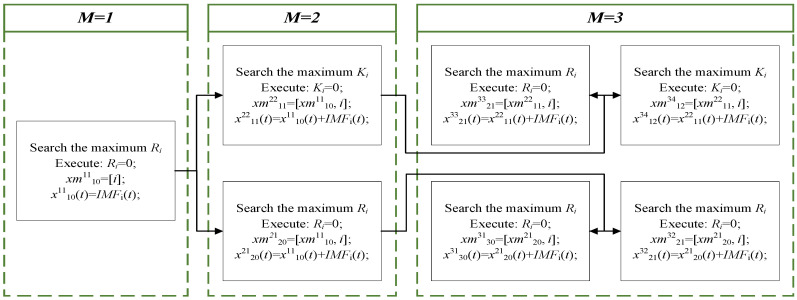
The selection rule of the effective IMFs.

**Figure 3 sensors-25-00913-f003:**
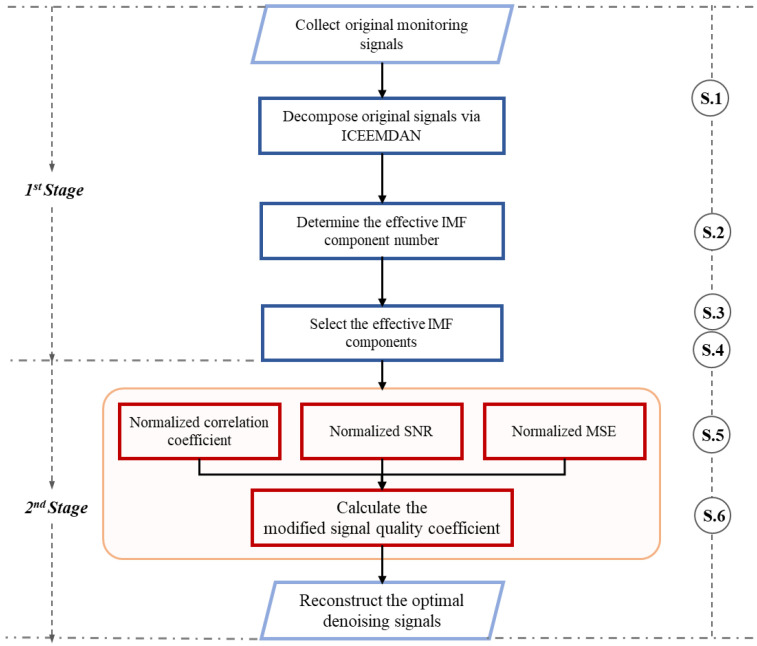
The flowchart of overall signal denoising processes based on MSQC.

**Figure 4 sensors-25-00913-f004:**
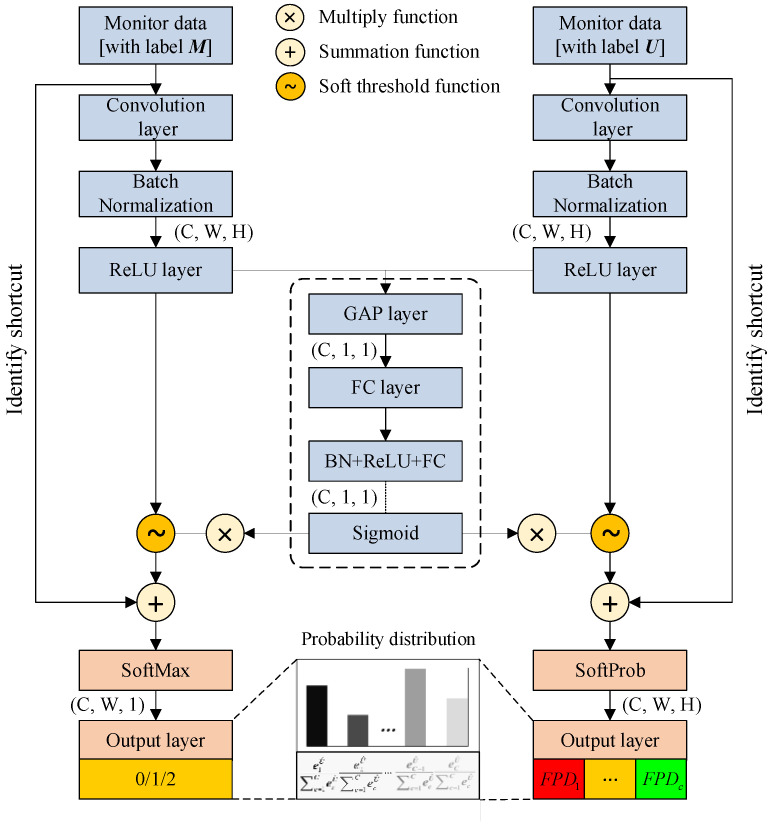
The architecture of versatile residual shrinkage network.

**Figure 5 sensors-25-00913-f005:**
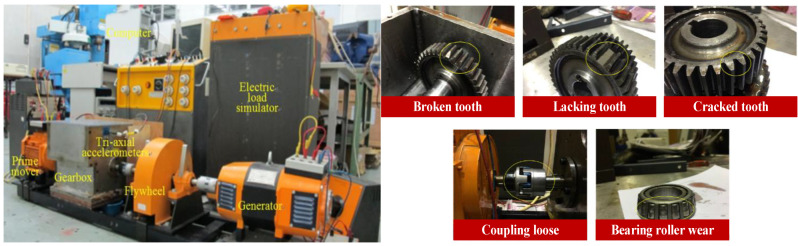
The self-built wind turbine gearbox compound fault test platform.

**Figure 6 sensors-25-00913-f006:**
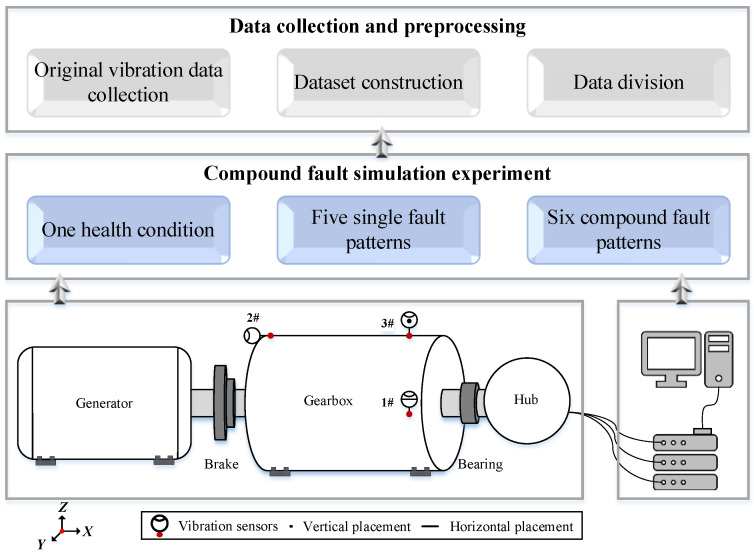
The overall schematic of the experimental study design.

**Figure 7 sensors-25-00913-f007:**
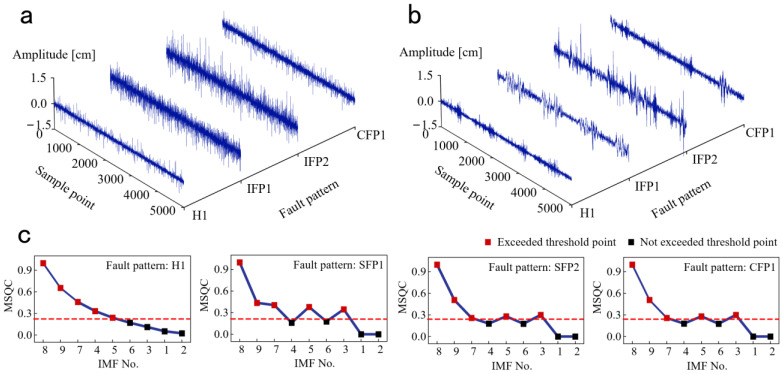
The denoised results of raw vibration signals under four fault patterns. (**a**) Raw vibration signals. (**b**) Denoised signals. (**c**) The MSQC values.

**Figure 8 sensors-25-00913-f008:**
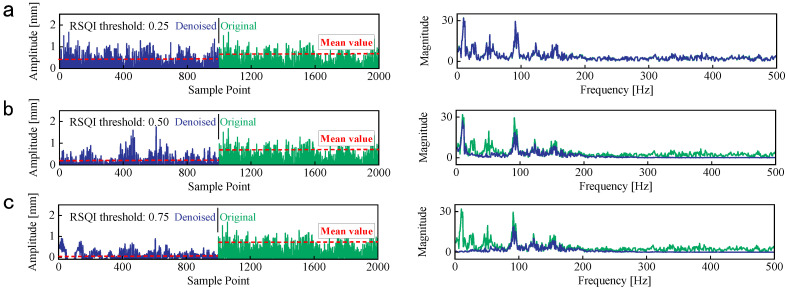
The denoised performance of CFP1 with four MSQC thresholds. (**a**) The denoised results with MSQC threshold = 0.25. (**b**) The denoised results with MSQC threshold = 0.5. (**c**) The denoised results with MSQC threshold = 0.75.

**Figure 9 sensors-25-00913-f009:**
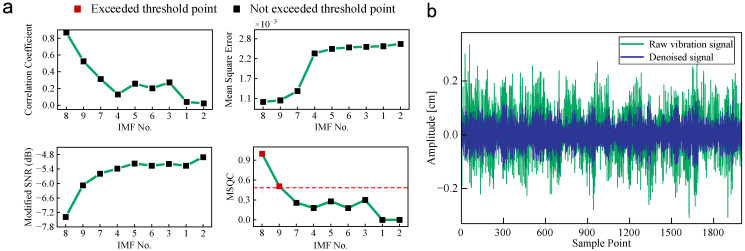
The denoised performance of CFP1 with the MSQC threshold = 0.5. (**a**) Correlation coefficient, mean square error, modified SNR and MSQC. (**b**) The raw vibration and denoised signals.

**Figure 10 sensors-25-00913-f010:**
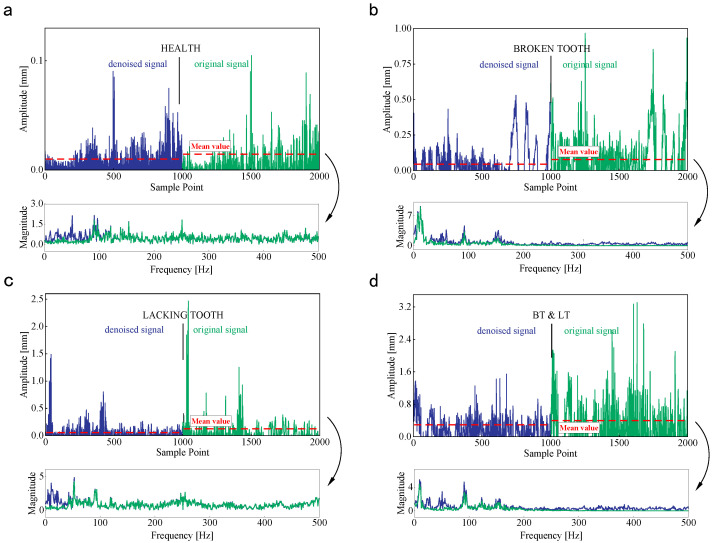
The original and denoised signals under four fault patterns. (**a**) H1. (**b**) IFP1. (**c**) IFP2. (**d**) CFP1.

**Figure 11 sensors-25-00913-f011:**
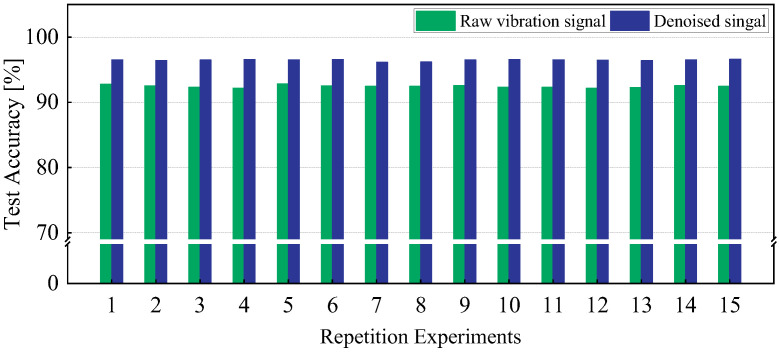
The test accuracy of fifteen repeated experiments before and after signal denoising.

**Figure 12 sensors-25-00913-f012:**
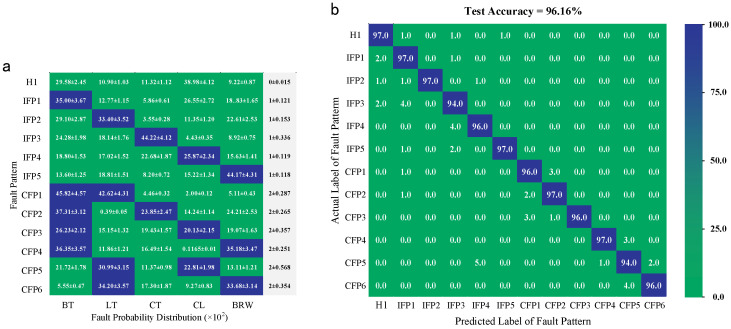
The diagnosis results after 15 repetition experiments. (**a**) The predicted fault probability distribution and fault number. (**b**) The multiclass confusion matrix of fault diagnosis for the test sets.

**Figure 13 sensors-25-00913-f013:**
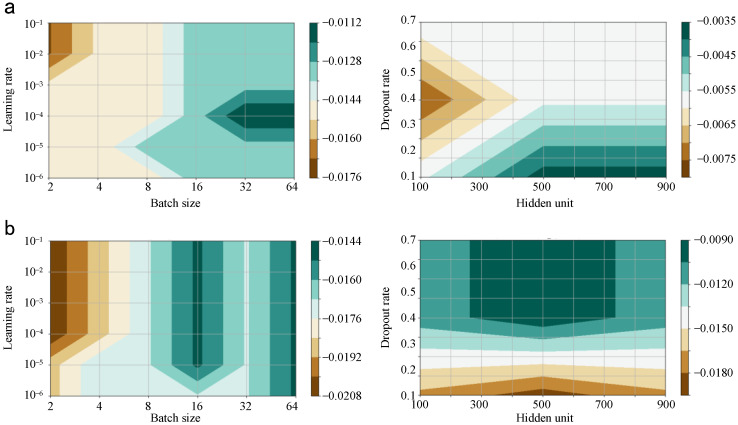
The parameters of Counter-DRSN and Locator-DRSN. (**a**) The cross-validation result of Counter-DRSN. (**b**) The cross-validation result of Locator-DRSN.

**Table 1 sensors-25-00913-t001:** A detailed description of fault number and category label.

Fault Pattern	Fault Description	Fault Label	Count Label	Training/Validation/Testing Number
H1	Health (H1)	0	0	1050/150/300
SFP1	Broken tooth (BT)	1	1	1050/150/300
SFP2	Lacking tooth (LT)	2	1	1050/150/300
SFP3	Cracked tooth (CT)	3	1	1050/150/300
SFP4	Coupling loose (CL)	4	1	1050/150/300
SFP5	Bearing roller wear (BRW)	5	1	1050/150/300
CFP1	BT and LT	1, 2	2	1050/150/300
CFP2	BT and CT	1, 3	2	1050/150/300
CFP3	BT and CL	1, 4	2	1050/150/300
CFP4	BT and BRW	1, 5	2	1050/150/300
CFP5	LT and CL	2, 4	2	1050/150/300
CFP6	LT and BRW	2,5	2	1050/150/300

**Table 2 sensors-25-00913-t002:** The compound fault diagnosis performances of seven methods.

Method No.	Diagnosis Strategies	Running Time (s)	Average Accuracy ± Standard deviation (%)		
			Training Set	Validation Set	Testing Set
Method 1 [8]	PPMLC	57.95	86.98 ± 6.24	86.12 ± 6.78	86.03 ± 6.89
Method 2	PPMLC with MSQC	75.34	89.87 ± 5.12	88.54 ± 5.64	88.32 ± 5.79
Method 3 [13]	RAKEL	81.34	82.34 ± 7.81	81.83 ± 8.12	81.54 ± 8.34
Method 4	RAKEL with MSQC	102.54	86.73 ± 6.875	84.57 ± 7.51	84.53 ± 7.53
Method 5 [6]	Dual-ELM	21.75	83.64 ± 7.84	82.57 ± 7.96	82.53 ± 7.99
Method 6	Dual-ELM with MSQC	39.57	87.51 ± 6.57	86.54 ± 6.89	85.98 ± 7.23
Method 7 [27]	WT-MLCNN	207.24	93.76 ± 3.251	92.87 ± 3.427	92.72 ± 3.457
Method 8	WT-MLCNN with MSQC	231.25	98.37 ± 1.387	95.81 ± 2.101	95.83 ± 2.024
Method 9 (Ours)	The proposed method	52.34	98.54 ± 1.360	96.34 ± 1.785	96.26 ± 1.823

## Data Availability

Data are available on request from the authors.

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
