# Peer review of "Compound Fault Diagnosis of Wind Turbine Gearbox via Modified Signal Quality Coefficient and Versatile Residual Shrinkage Network"

_sensors, 2025, doi:10.3390/s25030913_

Round 1
Reviewer 1 Report
Comments and Suggestions for Authors
1、 The quality of the figures must be improved to enhance readability, particularly Figure 1.
2、 The fault pattern name in Figure 10 differs from the definition provided in Table 1, please check.
3、 If the proposed model can count the number of faults, it should also be capable of identifying whether a fault is single or compound. Why does the residual shrinkage network require two branches? This reasoning must be explicitly and clearly explained.
4、 Collecting sensitive data and removing noise components hidden in the signals can be considered a physics-knowledge-guided method for fault diagnosis. Is there a connection or potential research direction similar to what was presented in the article “Physics-guided degradation trajectory modeling for remaining useful life prediction of rolling bearings”? It is recommended to discuss this in the introduction section to improve the paper's completeness.
Comments on the Quality of English Language
It can be published as is, but further improvements in language could be considered
Author Response
Thank you very much for taking the time to review this manuscript. Please find the point-to-point responses and the corresponding revisions in the attachment below.

Reviewer 2 Report
Comments and Suggestions for Authors
In the paper under review, the authors described their proposals for diagnosing wind turbine gearbox failure. This problem is not new; many researchers from various scientific teams and companies are constantly working on its solution. In the present study, the authors proposed a new method for diagnosing faults. The main idea of the method is the implementation of preliminary detection of removal of the noise components that are not related to the state of the wind turbine gearbox using the modified signal quality coefficient (MSQC) and further diagnostics using the versatile residual shrinkage network (VRSN). This undoubtedly has practical significance and will be of interest to scientists and researchers, specialists in the field of design and operation of wind turbine systems.
The research was carried out by simulation using the Matlab and Python softwares, as well as experimental studies using a specially created testing platform.
After reading the paper, I consider it necessary to note that:
1. The title of the paper corresponds to the research area.
2. The Abstract corresponds to the content of the paper.
3. The Introduction is comprehensive. It presents a literature review, where the prerequisites for conducting research are considered, the relevance of the study is explained, the purpose is formulated, and an analysis of studies previously published by other authors is given.
4. The number of references and their relevance seems sufficient. Of the 27 references, 13 are to articles published in the last 5 years. All references are provided appropriately.
5. The tables, figures and equations are designed in accordance with the requirements and have a sufficient description in the text.
6. The Conclusions are generally consistent with the presented evidence and arguments.
As recommendations that would improve the paper, I consider it necessary to note the following:
1. The theoretical background presented by the authors in the second section is too meager. It seems to me that the author should expand the presentation of the theoretical aspect of their proposals.
2. I ask the authors to clearly formulate the scientific novelty of the proposed solutions. In its present form, the paper looks like the use of known methods and algorithms in practical application to solving the problem.
3. In continuation of the previous comment, I suggest that in the third section, the authors not only describe the well-known MSQC and VRSN methods they applied but also emphasize their original solutions.
4. In my opinion, starting sections 3, 4, 5.1, 5.2 and 5.3 with a figure or table is a bad decision. I believe that:
- before Figure 1 it is necessary to provide its description;
- Figure 4 and Table 1 should be moved and positioned after lines 219-227;
- Figure 5 should be moved and positioned after lines 240-248;
- Figure 9 should be moved and positioned after lines 288-294;
In addition, I suggest the authors move and position:
- Figure 2 and Table 1 after lines 219-227;
- Figure 6 after lines 256-265;
- Figure 7 after lines 269-275;
- Figure 8 after lines 276-284.
In general, I can recommend this paper for publication after correcting the above shortcomings.
Author Response

(The authors gave the same response as above.)

Reviewer 3 Report
Comments and Suggestions for Authors
The paper addresses the diagnosis of faults in wind turbine gearboxes, which is vital for their efficiency and safety. Current methods often struggle with noisy signals and complex fault patterns. The authors propose a new approach using a modified signal quality coefficient (MSQC) to filter out irrelevant noise while preserving the signal's integrity and a versatile residual shrinkage network (VRSN) that counts faults and determines if they are single or multiple. This self-adaptive network eliminates the need for manual threshold adjustments and can perform various tasks simultaneously for real-time diagnosis. Testing on a custom-built rig for gearbox faults shows that this method is effective and accurate in diagnosing complex issues.
The work is interesting, but some adjustments are needed. Here are the comments:
1. I suggest adding a graphical abstract to show the study's experimental design.
2. To improve the manuscript's clarity and readability, focus on the following:
- Consistency in Abbreviations: Use abbreviations consistently throughout the text.
- Acronyms: Spell out acronyms on their first use, followed by the abbreviation in parentheses.
- Proofread for Typos and Formatting Issues: Check for minor spelling and formatting inconsistencies.
- Parallel Structure: Ensure grammatical consistency when listing items or ideas.
- Shorten Long Sentences: Break up long sentences for better readability.
3. Be careful when integrating figures into the text. Explaining the figure before inserting it into the text would be desirable. For example, Figure 1 is placed directly after the title, and you refer to it in section 3.1.
4. A flowchart for the steps presented in section 3.1 would be useful
5. Step 4, shown on page 4, lines 143-153, is a bit cumbersome. A good idea is to include Figure 2 after the paragraph ends.
6. Section 5.3 is well done, but I think the table should be placed at the end of this section.
Author Response

(The authors gave the same response as above.)

Round 2
Reviewer 2 Report
Comments and Suggestions for Authors
The authors provided answers to my comments and made necessary additions to the paper. I think that in this form the paper can be recommended for publication.